# MKOR: Momentum-Enabled Kronecker-Factor-Based Optimizer Using Rank-1 Updates

**Mohammad Mozaffari**
Department of Computer Science
University of Toronto
mmozaffari@cs.toronto.edu

**Sikan Li**
Texas Advanced Computing Server
sli@tacc.utexas.edu

**Zhao Zhang**
Department of Electrical and Computer Engineering
Rutgers University
zhao.zhang@rutgers.edu

**Maryam Mehri Dehnavi**
Department of Computer Science
University of Toronto
mmehride@cs.toronto.edu

## Abstract

This work proposes a Momentum-Enabled Kronecker-Factor-Based Optimizer Using Rank-1 Updates, called MKOR, that improves the training time and convergence properties of deep neural networks (DNNs). Second-order techniques, while enjoying higher convergence rates vs first-order counterparts, have cubic complexity with respect to either the model size and/or the training batch size. Therefore, they exhibit poor scalability and performance in transformer models, e.g. large language models (LLMs), because the batch sizes in these models scale by the attention mechanism sequence length, leading to large model size and batch sizes. MKOR's complexity is quadratic with respect to the model size, alleviating the computation bottlenecks in second-order methods. Because of their high computation complexity, state-of-the-art implementations of second-order methods can only afford to update the second order information infrequently, and thus do not fully exploit the promise of better convergence from these updates. By reducing the communication complexity of the second-order updates, as well as achieving a linear communication complexity, MKOR increases the frequency of second-order updates. We also propose a hybrid version of MKOR (called MKOR-H) that mid-training falls backs to a first order optimizer if the second order updates no longer accelerate convergence. Our experiments show that MKOR outperforms state-of-the-art first-order methods, e.g. the LAMB optimizer, and best implementations of second-order methods, i.e. KAISA/KFAC, up to $2.57\times$ and $1.85\times$ respectively on BERT-Large-Uncased on 64 GPUs.

## 1 Introduction

Second-order optimization methods have recently gained popularity in the training process of deep neural networks (DNNs) due to their higher convergence rate in comparison to their first-order counterparts. One of the well-known second-order methods, Newton's method, uses the inverse of the Hessian of the objective function as a preconditioner to the gradients, capturing more information on the curvature of the loss function. However, since the size of the Hessian scales with the model size, computing and storing the exact Hessian and its inverse is the main bottleneck in these methods, giving rise to different approximation methods.

One of the most common approximations of Newton's method is Natural Gradient Descent (NGD) [1], where the Hessian is substituted with the Fisher Information Matrix (FIM) [4] to reduce the cost

37th Conference on Neural Information Processing Systems (NeurIPS 2023).

of computing the second-order derivatives. However, as the models become larger, it becomes impractical to store and compute the exact inverse of the FIM, leading to the design of algorithms based on block-diagonal approximations of FIM; each block corresponds to a layer in the neural network (NN). Inverting the diagonal blocks is also not practical due to the large number of parameters in a NN layer, thus their inverse is also approximated.

The class of Kronecker-Factored Approximate Curvature (KFAC) [16] methods attempt to reduce the computation costs of the the block inversion. They approximate an FIM block for a batch of samples using the Kronecker multiplication of the covariance of the output activations and input gradients. KFAC is implemented on distributed platforms for both linear and convolutional layers [5; 27; 20; 19], and different computation and communication optimization techniques have been applied to its implementations [25; 21]. KAISA [21] is a framework with state-of-the-art distributed KFAC implementation. The computational complexity of KFAC-based methods is $\mathcal{O}(d^3)$, where $d$ is the layer dimension. These methods work well for small models with a small layer dimension $d$, however, in large models, they don't scale well, resulting in poor performance.

To reduce the effect of model size on the computation complexity of second-order updates, KBFGS [8], which does not have an efficient distributed implementation, uses a Broyden-Fletcher-Goldfarb-Shannon (BFGS) [13]-based method for computing and updating the Kronecker factor and has $\mathcal{O}(bd^2)$ complexity, where $b$ is the batch size. The class of SNGD (Sherman-Morrison-Woodbury-based NGD) methods [23; 31; 17] uses the Sherman-Morrison-Woodbury (SMW) identity to calculate the FIM blocks with a complexity of $\mathcal{O}(b^3)$, making the complexity independent of the model size. HyLo [17] is the state-of-the-art SNGD implementation. It reduces communication for better scalability. KBFGS and SNGD solve the scalability issue of KFAC-based methods for small batches. However, in transformer models [28], the batch size scales with the sequence length of the attention mechanism (which can be as high as a few thousands [26]) thus limiting the scalability of SNGD and KBFGS methods. Recent work Eva [33] attempts to reduce this cost further to $\mathcal{O}(d^2)$ by storing vectors instead of Kronecker factors in the KFAC formulation. However, similar to KFAC, Eva uses a damping factor that can lead to additional error in the FIM approximation. Also, because Eva stores the Kronecker vectors instead of factors, it can not leverage the benefits of momentum.

This work presents MKOR, a **M**omentum-Enabled **K**ronecker-Factorizatoin-Based **O**ptimizer with **R**ank-1 Updates. *(1)* MKOR approximates the inverse of covariance matrices using rank-1 updates in the *Sherman-Morrison-Based (SM-Based) Matrix Inversion*, reducing the inversion computation complexity from $\mathcal{O}(d^3)$ to $\mathcal{O}(d^2)$. As a result, the second-order information updates can be applied up to 100 times more frequently compared to KAISA/KFAC and HyLo/SNGD. KFAC computes and inverts the covariance matrices precisely; KFAC updates the second-order information infrequently, e.g. every 100-1000 iterations, due to the high cost of inversion and for time efficiency, which damages its convergence rate and generalization capability. SNGD-based methods suffer from similar overhead due to inversion of their kernel matrices. *(2)* Second-order methods suffer from high communication costs for synchronizing the inverse of factors between workers. MKOR alleviates this by only synchronizing rank-1 approximation vectors among the workers, reducing the communication costs from $\mathcal{O}(d^2)$ to $\mathcal{O}(d)$. Also, MKOR uses half-precision to further reduce communication; other second-order methods cannot use low-precision computations due to their complex matrix inversion algorithms. *(3)* Second-order methods are more prone to the exploding gradients due to the effects of preconditioning on the norm of the gradients and lack of numerical bounds on the inverse of the factors. MKOR uses a *Norm-Based Stabilizer* and a *Gradient Rescaling Mechanism* to detect and prevent exploding gradients. *(4)* The high convergence rate of second-order methods, including MKOR, stagnates after the first few iterations or epochs of training. We propose a hybrid version of MKOR, namely MKOR-H, that combines the high convergence rate of second-order methods in the initial phase of training with the low overhead of first-order methods in the late stages of the training, using a loss-reduction-rate-based switching mechanism.

MKOR outperforms state-of-the-art distributed second- and first-order methods by up to $2.57\times$, reducing the training time of BERT-Large-Uncased from 8 hours to 3 hours on 64 A100 GPUs. MKOR also achieves new state-of-the-art metrics on the GLUE dataset, where other second-order methods such as KFAC fail to converge to the baseline.

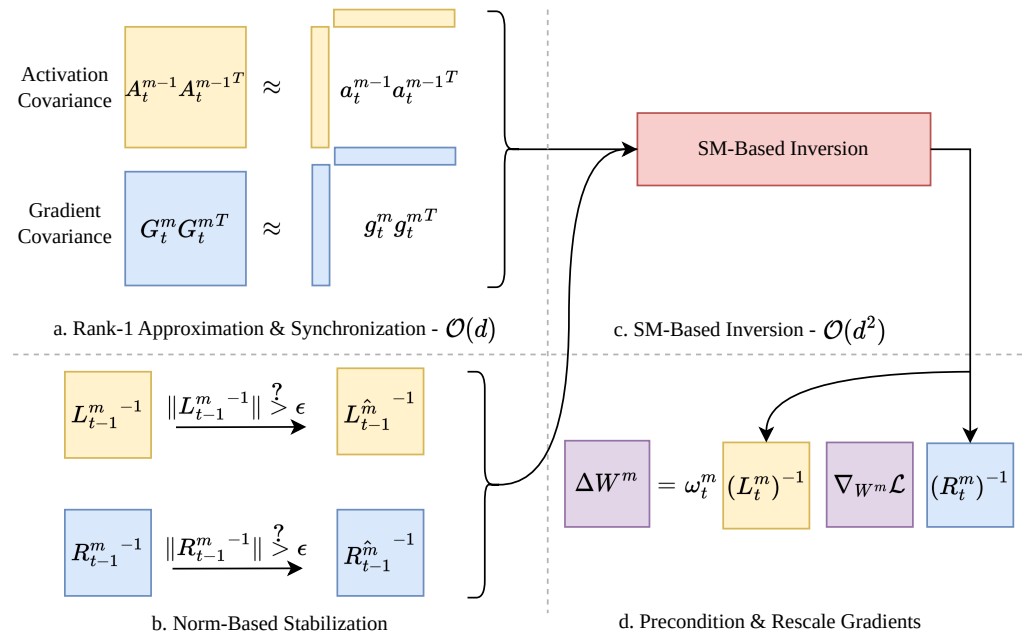

Figure 1: MKOR for layer $m$ on a single worker. The inputs of MKOR are the activations $A_t^m$, the gradients of the loss function with respect to the inputs $G_t^m$, and the gradients of the loss function with respect to the weights $\nabla_{W^m}\mathcal{L}$. The output is the update values $\Delta W^m$.

## 2 Background

Training a neural network involves solving an optimization problem to find the optimal values for a set of weights $\mathcal{W} = \{W^m\}_{m=1}^M$, where $M$ is the number of layers in the network and $W^m$ is a matrix in $\mathbb{R}^{d \times d}$. Second-order methods precondition the weights of the network with the inverse of the Hessian for better convergence rates. Block-diagonal approximations of NGD methods replace the Hessian with the block-diagonal FIM as shown in Equation 1, where $w^m \in \mathbb{R}^{d^2}$ is the vector representation of $W^m$, $F^m$ is the block corresponding to that layer and $\mathcal{L}$ is the loss function. [15] Shows that the FIM matches the Gauss-Newton matrix under certain conditions.

$$w^m := w^m - \alpha(F^m)^{-1}\nabla_{w^m}\mathcal{L} \tag{1}$$

KFAC-based methods reformulate the FIM block as the Kronecker product of two matrices. Equation 2 shows the update rule in KFAC, where $\mathcal{L}$ is the loss function and $(L_t^m)^{-1}$ and $(R_t^m)^{-1}$ are the inverse of the left and right factors, respectively.

$$W^m := W^m - \alpha(L_t^m)^{-1}\nabla_{W^m}\mathcal{L}(R_t^m)^{-1} \tag{2}$$

$(L_t^m)^{-1}$ and $(R_t^m)^{-1}$ in Equation 2 are computed using equations 3 and 4, respectively, where $a^m$ is the activation value of a sample at layer $m$, and $g_m = \nabla_{a^{m-1}}\mathcal{L}$ and $\gamma$ incorporate the momentum feature to avoid extreme changes in the factors.

$$L_t^m = \gamma L_{t-1}^m + (1 - \gamma)\mathbb{E}[g_t^m g_t^{mT}] \tag{3}$$

$$R_t^m = \gamma R_{t-1}^m + (1 - \gamma)\mathbb{E}[a_t^{m-1} a_t^{m-1T}] \tag{4}$$

## 3 MKOR: Momentum-Enabled Kronecker-Factorization-Based Optimizer with Rank-1 Updates

In this section we first present the MKOR algorithm, its computation and communication complexity, then present hybrid MKOR (MKOR-H), and finally discuss MKOR's convergence and stability.

---

**Algorithm 1:** MKOR Algorithm for a Single Layer $m$

---

**Data:** $A_t^{m-1}, G_t^m, W_{t-1}^m$

**Result:** $W_t^m$

1 **if** $m \in$ *Second Order Layers* **then**

2 $\quad \mathbf{a_t^{m-1}} \leftarrow \frac{1}{b}\sum_{i=1}^b (A_t^{m-1})_{:,i}$ ;      // Approx: $\quad A_t^{m-1}A_t^{m-1^T} \approx \mathbf{a_t^{m-1}}\mathbf{a_t^{m-1}}^T$

3 $\quad \mathbf{g_t^m} \leftarrow \frac{1}{b}\sum_{i=1}^b (G_t^m)_{:,i}$ ;          // Approx: $\quad G_t^m G_t^{m^T} \approx \mathbf{g_t^m}\mathbf{g_t^m}^T$

4 $\quad \mathbf{a_t^{m-1}}, \mathbf{g_t^m} \leftarrow$ AllReduce$(\mathbf{a_t^{m-1}}, \mathbf{g_t^m})$ ;     // Synchronize Approximations

$\quad$ // Norm-Based Stabilization

5 $\quad L_{t-1}^{\hat{m}}{}^{-1} \leftarrow$ **if** $\|L_{t-1}^m{}^{-1}\| > \epsilon$ **then** $\zeta L_{t-1}^m{}^{-1} + (1-\zeta)I$ **else** $L_{t-1}^m{}^{-1}$ ;

6 $\quad R_{t-1}^{\hat{m}}{}^{-1} \leftarrow$ **if** $\|R_{t-1}^m{}^{-1}\| > \epsilon$ **then** $\zeta R_{t-1}^m{}^{-1} + (1-\zeta)I$ **else** $R_{t-1}^m{}^{-1}$ ;

$\quad$ // SM-Based Factor Inversion

7 $\quad L_t^{m-1} \leftarrow \gamma L_{t-1}^{\hat{m}}{}^{-1} + \frac{(1-\gamma)}{\gamma^2(1+\gamma(1-\gamma)\mathbf{g_t^m}^T L_{t-1}^{\hat{m}}{}^{-1}\mathbf{g_t^m})} L_{t-1}^{\hat{m}}{}^{-1}\mathbf{g_t^m}\mathbf{g_t^m}^T L_{t-1}^{\hat{m}}{}^{-1}$

8 $\quad R_t^{m-1} = \gamma R_{t-1}^{\hat{m}}{}^{-1} + \frac{(1-\gamma)}{\gamma^2(1+\gamma(1-\gamma)\mathbf{a_t^m}^T R_{t-1}^{\hat{m}}{}^{-1}\mathbf{a_t^m})} R_{t-1}^{\hat{m}}{}^{-1}\mathbf{a_t^m}\mathbf{a_t^m}^T R_{t-1}^{\hat{m}}{}^{-1}$

9 $\quad \Delta \hat{W}_t^m \leftarrow L_t^{m-1}\nabla_{W^m}\mathcal{L}R_t^{m-1}$ ;          // Precondition Gradients

10 $\quad \Delta W_t^m \leftarrow \frac{\|\nabla_{W^m}\mathcal{L}\|}{\Delta \hat{W}_t^m}\Delta \hat{W}_t^m$ ;          // Rescale Gradients

11 **else**

12 $\quad \Delta W_t^m \leftarrow \nabla_{W^m}\mathcal{L}$ ;

13 **end**

14 $W_t^m \leftarrow Optimizer.step(\Delta W_t^m, W_{t-1}^m)$ ;

---

### 3.1 The MKOR Algorithm

Algorithm 1 summarizes the MKOR optimizer for a single layer and Figure 1 shows the workflow. For each layer (line 1 in Algorithm 1) MKOR updates the second-order information and preconditions the gradients, and at the end the backend optimizer updates the weight using the preconditioned gradients (line 14 in Algorithm 1).

*Rank-1 Approximation.* For the rank-1 approximations of the covariance matrices, we use the average of the values across all the samples, i.e. $\mathbf{a_t^{m-1}} = \mathbb{E}[a_t^{m-1}]$ and $\mathbf{g_t^m} = \mathbb{E}[g_t^m]$ (lines 2 and 3 in Algorithm 1 and Figure 1-a). $(A_t^{m-1})_{:,i}^{-1}$ and $(G_t^m)_{:,i}^{-1}$ show the $i^{th}$ column of $(A_t^{m-1})^{-1}$ and $(G_t^m)^{-1}$ respectively, where $A_t^{m-1}$ and $G_t^m$ are the activations and the gradients of layer $m$ respectively.

*Norm-Based Stabilizer.* The values in the factor inverses in second-order methods can become large or vanish due to extremely large or small values in activations and gradients, leading to numerical instabilities and over/underflows. Since the inverse of the factors are directly multiplied by the gradients to find the update values, it can cause oscillations or even divergence. MKOR uses a norm-based stabilizer to detect this and addresses it by modifying the inverse of the factors accordingly (lines 5 and 6 in Algorithm 1 and Figure 1-b). More detail on the norm-based stabilizer are in Section3.3.

*SM-Based Inverter.* MKOR directly modifies the inverse of the left and right factors using rank-1 updates, while using the momentum for better convergence. If $\mathbb{E}[g^m g^{m^T}]$ is approximated using a rank-1 matrix $\mathbf{g^m}\mathbf{g^{m^T}}$ and using the Sherman-Morrison identity, Equation 5 in obtained (line 7 in Algorithm 1 and Figure 1-c).

$$L_t^{m-1} = \gamma L_{t-1}^m{}^{-1} + \frac{(1-\gamma)}{\gamma^2(1+\gamma(1-\gamma)\mathbf{g_t^m}^T L_{t-1}^m{}^{-1}\mathbf{g_t^m})}L_{t-1}^m{}^{-1}\mathbf{g_t^m}\mathbf{g_t^m}^T L_{t-1}^m{}^{-1} \quad (5)$$

Furthermore, if Equation 4 is approximated using $\mathbb{E}[a_t^{m-1}a_t^{m-1^T}] \approx \mathbf{a_t^m}\mathbf{a_t^{m^T}}$ with a similar derivation, Equation 6 is obtained (line 8 in Algorithm 1 and Figure 1-c).

Table 1: The computation and communication complexity and memory overhead of the state-of-the-art implementations of the first- and second-order optimizers. The division by 2 in MKOR is because MKOR uses half-precision computations. The complexity of KFAC-based methods depends on layer dimensions while SNGD methods mostly depend on the batch size. In transformers, due to the scaling of the batch size by the sequence length, batch sizes and layer dimensions are comparable, making both KFAC- and SNGD-based methods more expensive than SGD.

| Optimizer | Computational Complexity | Memory Overhead | Communication Complexity |
|---|---|---|---|
| MKOR | $\mathcal{O}(d^2 + bd)$ | $\mathcal{O}(2d^2/2)$ | $\mathcal{O}(2d/2)$ |
| SNGD (HyLo) | $\mathcal{O}(b^3)$ | $\mathcal{O}(2bd + b^2)$ | $\mathcal{O}(2bd + b^2)$ |
| KFAC (KAISA) | $\mathcal{O}(d^3)$ | $\mathcal{O}(4d^2)$ | $\mathcal{O}(4d^2)$ |
| Eva | $\mathcal{O}(d^2 + bd)$ | $\mathcal{O}(2d)$ | $\mathcal{O}(2d)$ |
| SGD (Momentum) | - | $\mathcal{O}(d^2)$ | - |
| ADAM / LAMB | - | $\mathcal{O}(d^2)$ | - |

$$R_t^{m-1} = \gamma R_{t-1}^{m}{}^{-1} + \frac{(1-\gamma)}{\gamma^2(1 + \gamma(1-\gamma)\mathbf{a_t^m}^T R_{t-1}^{m}{}^{-1}\mathbf{a_t^m})} R_{t-1}^{m}{}^{-1}\mathbf{a_t^m}\mathbf{a_t^m}^T R_{t-1}^{m}{}^{-1} \tag{6}$$

*Rescaling Gradients.* Preconditioning the gradients using the computed factors can change gradient norms. Sometimes, these changes interfere with the effect of the learning rate on the training process. To alleviate this and to make learning rate schedulers more effective, the preconditioned gradients are scaled so that their norm matches the original norms (line 10 in Algorithm 1 and Figure 1-d).

**Complexity Analysis.** MKOR reduces the memory, communication, and computation costs for factor inversion. Table 1 compares the overheads of different optimizers. *(1) Computation Complexity.* MKOR inverts the left and right factors in Equation 2 using equations 5 and 6, both of which can be computed using matrix-vector multiplications, and have $\mathcal{O}(d^2)$ computation complexity, in contrast to KFAC and SNGD methods that need $\mathcal{O}(d^3)$ and $\mathcal{O}(b^3)$ complexity to invert matrices in $\mathbb{R}^{d \times d}$ and $\mathbb{R}^{b \times b}$ respectively. *(2) Communication Complexity.* The only data that is synchronized among different workers in MKOR is the two rank-1 approximations that have $2d$ elements. With quantization, this size can be halved. In KFAC, the activation and gradient covariance matrices and the inversion of left and right factors need to be synchronized between all the workers, leading to $4d^2$ data transfers. In SNGD , the activations and gradients are synchronized, leading to $2bd$ data transfers and the inverted kernels are broadcast, resulting in $b^2$ data transfers. Reducing the communication complexity of MKOR from quadratic to linear results in better performance on large number of workers. *(3) Memory Overhead.* MKOR needs to store the inverse of the left and right factors and two rank-1 approximation vectors, leading to $2d^2 + 2d$ memory overhead, and using half-precision computations further reduces this. KFAC stores the activation and gradient covariance matrices and the left and right factors, leading to $4d^2$ memory overhead. SNGD stores the activations, the gradients, and the kernels they use as second-order information, leading to $2bd + b^2$ memory complexity.

The low memory overhead of MKOR allows us to use larger batch sizes compared to other second-order methods. In practice, gradient accumulation methods are used to increase the effective batch size in training, which reduces the training speed significantly. This issue worsens with KAISA and HyLo, but MKOR alleviates this. For fairness, in our experiments, we set the local batch sizes to the same value in all optimizers and do not leverage this feature of MKOR.

### 3.2 Hybrid MKOR

We observed that second-order methods, including MKOR, usually accelerate training more during the first iterations of the training time, and as the loss flattens, their advantage over their first-order counterparts becomes less noticeable. This is because the second-order information of the loss functions approach identity near convergence points. Thus we designed a hybrid second- and first-order optimizer with a loss decrease rate-based switching method (MKOR-H). MKOR-H evaluates the changes in the loss function in different iterations and switches back to first-order methods if needed for an efficient trade-off between the costly second-order updates and their benefits for convergence.

### 3.3 MKOR Convergence and Stability

**Inversion Frequency** Due to the high factor inversion costs in KFAC- and SNGD-based methods, researchers use the stale factor approach, which updates the inverted factors every $f$ iterations and reuses the results in the other iterations in their preconditioning to reduce the computation and communication costs. $f$ is the reciprocal of inversion frequency and varies from a few 100s to a few 1000s. Our experiments show that in average-sized models such as ResNet-50 [9], in an iteration that the inversion of factors is executed, the cost of KAISA and HyLo is $150\times$ more than an SGD iteration. Furthermore, more than 98% of the total cost in those iterations are spent on matrix inversion.

The stale factors approach can lead to good preconditioners if the loss function landscape does not vary significantly in each iteration. However, this is a strong assumption and doesn't necessarily hold in practice. Also, increasing the inversion frequency can benefit the convergence rate of the second-order methods. In addition, our experiments show that using stale factors. The stale factors can be good preconditioners lead to converging to local minima in the loss function and damage the generalization of the model.

**Numerical Stability** In second-order techniques, we need to invert or find the roots of matrices of different sizes, which are usually not full-rank, resulting in numerical issues. The KFAC implementation uses singular value decomposition (SVD) of the factors and masks the eigenvalues that are close to zero to deal with singular matrix inversion issues. In practice, the eigenvalues of the left and right factors in KFAC-based methods computed from equations 3 and 4 are increased manually by adding $\mu I$ to each of them to improve numerical stability ($\mu > 0$ is called the damping factor), but MKOR doesn't need such numerical fixes. Furthermore, HyLo uses two decomposition methods to sample the batch of inputs, namely KID and KIS. KID requires inverting matrices in $\mathbb{R}^{b \times b}$ of rank $min(b, d)$, thus for batch sizes larger than $d$ in a specific layer, the method fails.

Unlike SVD or other iterative methods used for factor inversion, MKOR doesn't suffer from numerical instabilities that rise from large condition numbers. MKOR has a single scalar division, in which the denominator is guaranteed to be non-zero based on lemma 3.1, eliminating the numerical over/under-flow possibility and the need for damping factors (required by other second-order methods for computational stability).

**Lemma 3.1.** *The factors computed using Equation 5 and 6 are all positive-definite.*

[2] suggests using double precision representation of numbers to avoid numerical over/under-flow in inverting or computing the roots of matrices. This approach adds more costs to the matrix inversion and increases the time complexity of the main bottleneck in second-order methods.

MKOR does not need higher precision computations, and can use half-precision floating point operations to reduce costs significantly. This will improve the memory utilization and reduce the communication costs in GPUs by $2\times$ while using cheaper computation blocks for half-precision operations. Lemma 3.2 shows an upper bound on the quantization error effect in the MKOR updates.

**Lemma 3.2.** *Assuming that the maximum quantization error is $\epsilon$, the maximum number in matrices and vectors is $m$, and the dimension of the vectors and matrices are $d$ and $d \times d$ respectively, the quantization error of formulas 5 and 6 is $O((\gamma + 4\frac{(1-\gamma)}{\gamma^2}m^3d^2)\epsilon)$*

**Exploding Gradients Problem** In second-order methods, where the gradients are preconditioned by various factors, the exploding gradient problem is worsened. Our experiments show that in first-order methods, by choosing a learning rate that doesn't lead to divergence in the first few iterations, explosion in gradients almost never occurs. On the other hand, in second-order methods, we observe that the explosion can occur at any iteration, and both KFAC and SNGD implementations are prone to this problem. This can lead to ripples in accuracy and divergence. One of the main approaches for solving the exploding gradient problem is choosing small values for the learning rate, limiting the convergence rate significantly. In particular, small learning rates damage the second-order methods and make them almost as performant as their first-order counterparts. Considering that SGD is more robust against the exploding gradients and taking advantage of the direct control of MKOR on the inverse of the factors, the factors in MKOR are modified to lean toward SGD once the possibility of exploding gradients is detected using equations 7 and 8, where $\zeta$ is a hyperparameter that controls the amount of information from the original factors that needs to be saved in the new factors.

$$\hat{L}_t^m = \zeta L_t^m + (1 - \zeta)I \qquad (7)$$

$$\hat{R_t^m} = \zeta R_t^m + (1 - \zeta)I \tag{8}$$

By expanding Equation 2 with the new factors, we will get Equation 9, which reduces the loss-based on lemma 3.3. The first term in the right-hand-side of 9 is the KFAC term, the second and third terms are the left and right preconditioned versions, and the last term is the SGD term.

$$\begin{aligned}
\hat{L}^{m^{-1}} \nabla_{W^m} \mathcal{L} \hat{R}^{m^{-1}} &= \zeta^2 L^{m^{-1}} \nabla_{W^m} \mathcal{L} R^{m^{-1}} \\
&+ \zeta(1-\zeta) L^{m^{-1}} \nabla_{W^m} \mathcal{L} + \zeta(1-\zeta) \nabla_{W^m} \mathcal{L} R^{m-1} + (1-\zeta)^2 \nabla_{W^m} \mathcal{L}
\end{aligned} \tag{9}$$

**Lemma 3.3.** *Given a differentiable function $\mathcal{L}(w)$ with first-order Taylor series approximation $\hat{\mathcal{L}}(w - \Delta w) = \mathcal{L}(w_0)\Delta w^T \nabla_w \mathcal{L}(w_0)$ around point $w_0$, assuming that at point $w_0$ the second-order derivative of the function $\mathcal{L}(w)$ is given as $\nabla_w^2 \mathcal{L}(w_0) = H = L \otimes R$, where $L$ and $R$ are positive-semi-definite matrices, for a value of $\Delta w = ((\zeta L^{-1} + (1-\zeta)I) \otimes (\zeta R^{-1} + (1-\zeta)I)) \nabla \mathcal{L}(w_0)$, the inequality $\hat{\mathcal{L}}(w_0 - \Delta w) < \mathcal{L}(w_0)$ holds.*

While this modification can avoid exploding gradients, overusing it with small values of $\zeta$ will convert MKOR to SGD. MKOR uses a factor norm-based metric that observes the infinity norm of the factors, and if they are greater than a specific threshold, the process of factor modification will be triggered.

## 4 Experimental Results

In this section, we demonstrate the performance of MKOR on a large language model using different benchmarks, and analyze the timing of different components in different first- and second-order algorithms. For results on more models and training sets, please refer to the supplementary material.

**Large Language Models.** We pre-train BERT-Large Uncased and fine-tune it for different question-answering and text classification tasks. A setup similar to [21] for pre-training and fine-tuning is used. The first-order baseline used is Fused LAMB [32]. Similar to [21], for the pre-training process, the English Wikipedia [30] and the Toronto BookCorpus [34] dataset, which was used in the original BERT pre-training, are used; the latter dataset is not thoroughly available which results in a small reduction in the baseline accuracies achieved in our experiments from the original BERT results. Following [21], due to the time-intensive process of hyperparameter tuning for the first phase of pre-training, we report the effectiveness of MKOR in the second phase of pre-training only while using the checkpoints of the first phase generated using LAMB optimizer.

As expected, the computation, communication, and memory complexity of HyLo is high, and the Khatri-Rao-based Interpolative Decomposition (KID) approximation method, the main idea of HyLo, cannot be executed because a single sample cannot fit into the 40GB memory of an A100 GPU. In addition, HyLo doesn't support gradient accumulation due to its memory complexity, depending on the batch size; in LLMs such as BERT, the batch sizes are as large as 64k.

For the question answering task, we fine-tune the pre-trained BERT checkpoints on the SQuAD v1.1 [22] dataset. Table 2 shows the F1 Score achieved using different optimizers and compares their convergence rate and speedups. The vanilla MKOR and KAISA both converge after 1000 iterations, while the LAMB optimizer requires $1,536$ steps. Considering that each step in MKOR is faster than KAISA, MKOR achieves an end-to-end speedup. MKOR-H will converge in 600 steps, reducing the number of steps in LAMB by $2.6\times$, while achieving the same accuracy. In addition, it achieves $2.57\times$ speedup over the LAMB optimizer and $1.75\times$ speedup over KAISA. As another second-order baseline, we consider Eva, which converges in 1000 iterations, and MKOR achieves $1.69\times$ speedup over it.

For classification tasks, we fine-tune BERT on the GLUE [29] dataset. Table 3 compares the results for different classification tasks in the GLUE dataset. MKOR with 1500 steps achieves a new state-of-the-art accuracy in GLUE dataset on BERT-Large Uncased, and MKOR and MKOR-H with 600 steps achieve the same average metric as the baseline, while reducing the number of steps by a factor of $2.6\times$. MKOR and MKOR-H both achieve $2.57\times$ end-to-end speedup. After training KAISA for $1,563$ steps, the model does not converge to the baseline average accuracy, while slowing down the convergence by $0.89\times$. Eva requires 1000 steps to converge to the target average metric, being $1.69\times$ slower than MKOR-H with 600 steps and $1.24\%$ less accurate than MKOR with 1500 steps.

Table 2: BERT-Large Uncased results on SQuAD v1.1 question answering task

| Metric | LAMB | KAISA | MKOR | MKOR-H | Eva |
|---|---|---|---|---|---|
| F1 | 90.44 | 90.44 | 90.50 | 90.64 | 90.55 |
| # Iterations | 1,536 | 1,000 | 1,000 | 600 | 1,000 |
| Time (h) | 7.97 | 5.71 | 5.25 | 3.10 | 5.24 |
| Speedup ($\times$) | 1.00 | 1.39 | 1.51 | 2.57 | 1.52 |

BERT-Large-Uncased Training Loss

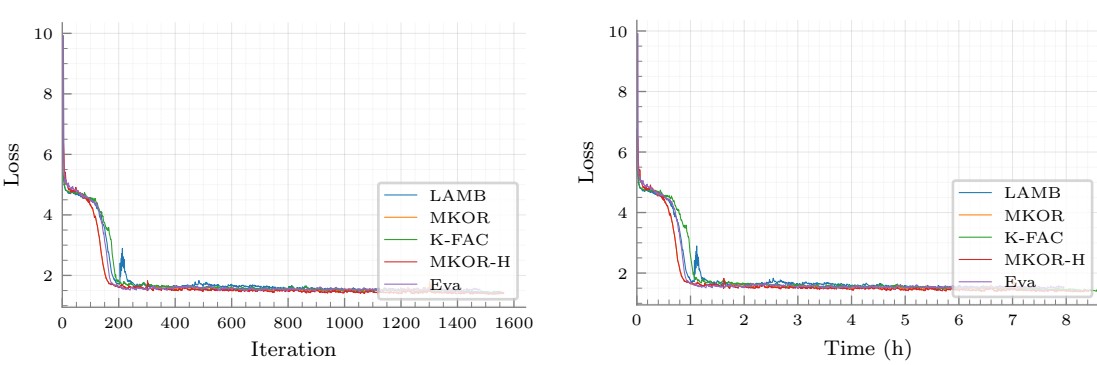

Figure 2: The training loss of BERT-Large-Uncased using different optimizers.

Per figure 2, which shows the training error during the training of BERT, MKOR decreases the error in less iterations in comparison to KAISA, Eva, and LAMB, leading to faster convergence. From Tables 2 and 3, MKOR-H converges in only 600 steps.

**Inversion Frequency.** Due to the low computation complexity of the updates on MKOR, the factor inversion frequency ($f$) in MKOR is in the range of 10. Figure 4-a shows that while the average iteration cost in KAISA is heavily dependent on the inversion frequency, MKOR's cost is almost independent of the inversion frequency. Also 4-b shows that increasing the inversion frequency leads to higher convergence rate. In addition, using stale factors may result in converging to a local minima. Hence, in MKOR we increase the convergence rate by updating the factors more frequently, without affecting the per-iteration cost, leading to end-to-end speedups in training. We use a simple autoencoder [24] on CIFAR-100 [11] in this experiment.

**Performance Analysis.** We compare the performance of different parts of the optimizers to illustrate the bottlenecks and advantages of different methods. The training process for an optimizer has three steps: factor computation, precondition, and update weights. Figure 3 shows the time spent on each task in different optimizers on two models; ResNet-50, a CNN and BERT-Large-Uncased, a transformer-based LLM with large sequence length. Since first-order optimizers such as SGD, ADAM, and LAMB don't require factorization and preconditioning, their optimization time is only spent in updating the weights. In ResNet-50, since the model size is larger compared to the batch size, the factor computation and inversion is more expensive for KAISA compared to HyLo. This cost is significantly reduced in MKOR.

For BERT-Large-Uncased, because of the large size of the model, the factor inversion time for KAISA is large. Also, due to the large sequence length value in this model, the kernel inversion time for

Table 3: BERT-Large Uncased results on the GLUE classification tasks.

| Optimizer | Iterations | Time (h) | Speedup ($\times$) | Average Metric |
|---|---|---|---|---|
| LAMB | 1,563 | 7.97 | 1.00 | 0.8023 |
| KAISA | 1,563 | 8.93 | 0.89 | 0.796 |
| MKOR | 1,500 | 7.88 | 1.01 | 0.8214 |
| MKOR | 600 | 3.10 | 2.57 | 0.8078 |
| MKOR-H | 600 | 3.10 | 2.57 | 0.811 |
| Eva | 1000 | 5.24 | 1.52 | 0.809 |

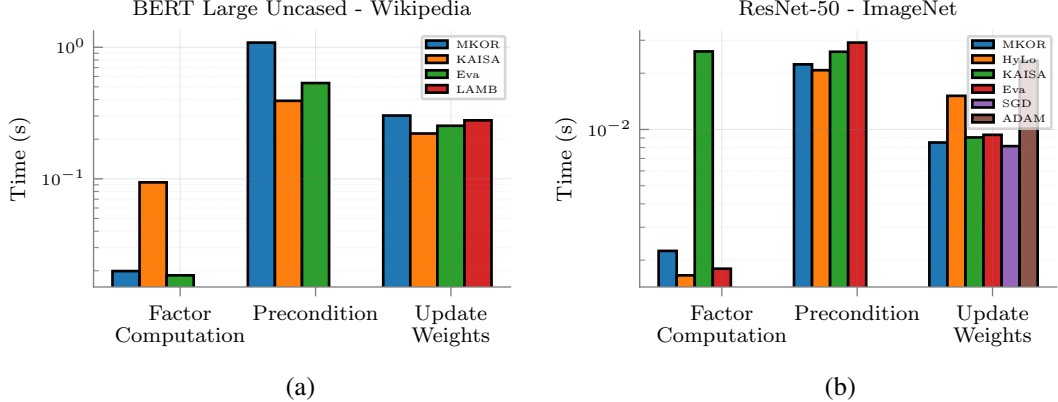

Figure 3: Per-step breakdown of different optimizes on BERT-Large-Uncased (**a**) and ResNet-50 (**b**)

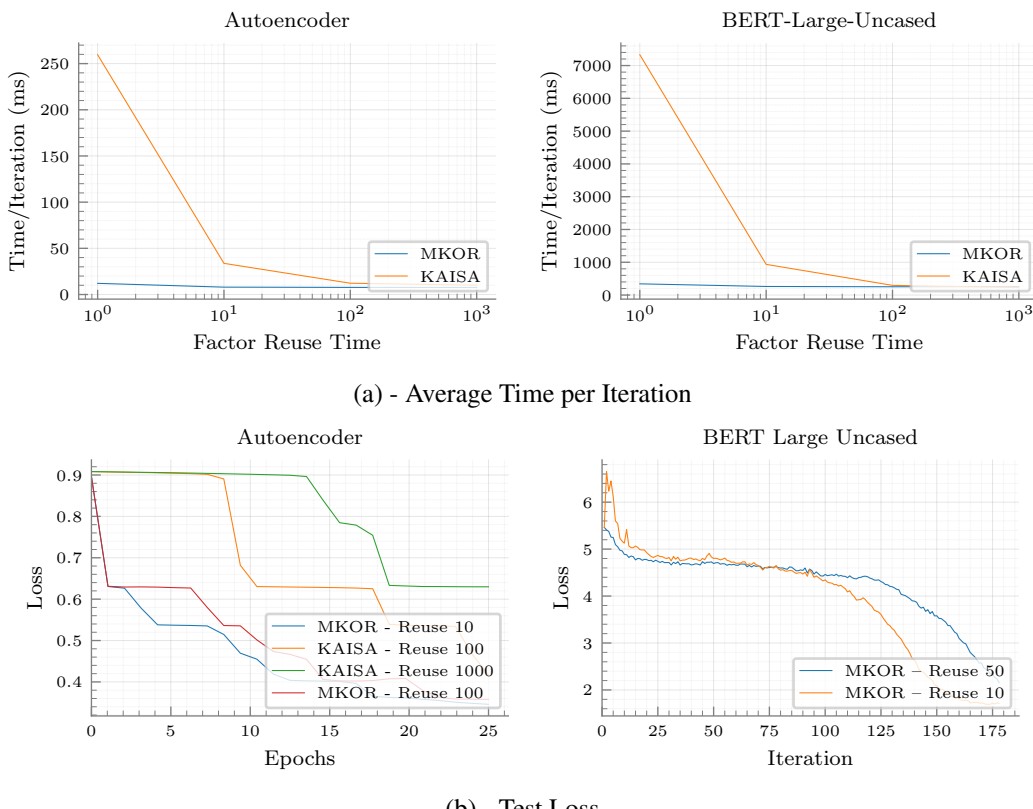

(a) - Average Time per Iteration

(b) - Test Loss

Figure 4: The sensitivity of MKOR and KAISA for BERT-Large-Uncased and an Autoencoder model (**a**) and the effect of inversion frequency on the convergence properties of these models (**b**).

HyLo is comparable to KAISA's inversion time. But as expected, because of its low computational complexity, the aforementioned cost in our method is much smaller than the total training time, leading to speedups. It is important to note that HyLo diverges in this training process, hence convergence time is not reported for HyLo.

**Approximation Error Experimental Results.** Due to the low-rank properties of the covariance matrices, MKOR utilizes rank-1 approximations of the covariance matrices to accelerate the computations and communication in KFAC-based optimizers. Here, we aim to theoretically and experimentally support this choice. As shown in Figure 5, our experiments show that the covariance matrices can be approximated with rank-1 matrices with low error and higher rank approximations are unnecessary in practice. Figure 5 shows the error distribution of the optimal rank-1 approximation methods of the covariance matrices in ResNet-50 and BERT-Large-Uncased pre-training. Our extensive tests

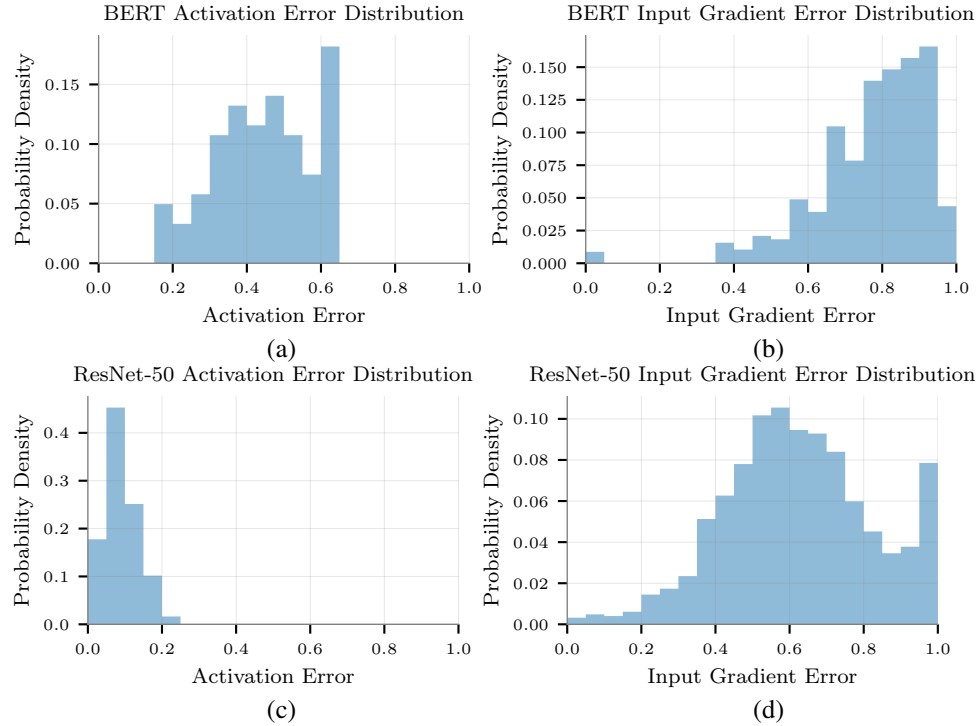

Figure 5: Rank-1 error for activation and input gradient covariance matrices for BERT-Large-Uncased pre-training (**a, b**) and ResNet-50 on ImageNet (**c, d**).

on well-known benchmarks show this property holds for all models and we have not come across a benchmark that does not have low-rank covariance matrices.

**Approximation Error Analysis and Extension to Higher Ranks.** Small batch sizes and over parameterization of networks will lead to low-rank covariance matrices in DNNs. Let's consider the covariance matrix $C = XX^T$, where $C \in R^{d \times d}$ is the covariance matrix and $X \in R^{d \times b}$ is a matrix in which each column corresponds to a single sample and $d$ and $b$ are the sample dimension and the per-GPU batch size respectively. Rank of the covariance matrix is $min(b, d)$. If the per-GPU batch sizes are small, the covariance matrices in each GPU will be low-rank. Rank-1 approximation methods can work well in these scenarios. If the batch sizes in each GPU are large, we observe that the covariance matrices will stay low-rank. The underlying reason for this observation is that current neural networks are over-parameterized, and as a result, different features in the covariance matrices of the activations and the output gradients won't be linearly independent, resulting in low-rank covariance matrices.

*Extending MKOR to Higher Ranks:* Furthermore, one can extend MKOR to use higher-rank covariance matrices. Let's assume that $C = \sum_{i=1}^{r} c_i c_i^T$ where $r$ is the rank of the covariance matrix $C$. We can apply SMW identity to compute $C_1^{new} = (C^{old} + c_1 c_1^T)^{-1}$ with $O(d^2)$ computational complexity. Then we can compute $C_2^{new} = (C_1^{new} + c_2 c_2^T)^{-1}$ using SMW identity with $O(d^2)$ computational complexity. We can continue the same pattern by computing $C_i^{new} = (C_{i-1}^{new} + c_i c_i^T)^{-1}$. The total computation complexity of this process will be $O(rd^2)$. We should add this cost to the cost of computing the low-rank approximation of $C$ which requires an SVD. Using SVD kills the main advantage of using low-rank computations, since the computational complexity of applying SVD is the same as inverting the factors directly. We could not find any cheaper way to compute low-rank approximations of the covariance matrices, except for the rank-1 approximation used in this paper.

## 5 Conclusion

We propose MKOR, a Momentum-Enabled Kronecker-Factor-Based Optimizer Using Rank-1 updates that improves the end-to-end training time and convergence rate of second-order methods by reducing their computation, communication, and memory usage complexity. Our experiments illustrate that MKOR outperforms state-of-the-art first- and second-order optimizers on large language models.

# 6 Broader Impact

The research described in this paper introduces a novel method for training DNNs that achieves faster convergence in LLMs. Using our work can help save a lot of energy and time for machine learning practitioners. The computations in this work are fully transparent. DNNs can be applied to different problems, including medical diagnostics, voice recognition, and addressing climate changes. However, it should be acknowledged that optimization algorithms for DNNs can also be applied to models with potentially negative implications such as privacy invasion. It is important to note that the responsible and ethical utilization of any efficient optimization algorithm, including those we propose, extends beyond the scope of this research.

# 7 Acknowledgments

We thank the reviewers for their constructive feedback. We would like to express our deepest appreciation to Dr. Behrooz Zarebavani for all the time and energy he put into helping with the writing and formatting of this paper. This work is supported by NSERC Discovery Grants (RGPIN06516, DGECR00303, RGPIN-2023-04897, DGECR-2023-00133), National Science Foundation (NSF CCF-2106621), the Canada Research Chairs program, the Ontario Early Researcher Award, the Digital Research Alliance of Canada (`https://www.alliancecan.ca`) and Texas Advanced Computing Center (`https://www.tacc.utexas.edu`). Work of Zhao Zhang was supported by OAC-2106661.

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

# 8 Supplementary Material

In this section, we start by discussing an experiment on training the ResNet-50 [9] model on ImageNet [7] dataset and compare MKOR against state-of-the-art second-order and first-order baselines. Then, for completeness, we report the GLUE results achieved on each task in 8.2. Next, we discuss the derivation of KFAC and SNGD approximation methods in 8.3. We then discuss some of the features of MKOR an other optimizers and back them up with quantitative data in 8.4 and 8.5. We will provide scalability results of MKOR in 8.6 and provide more data to back up the low-rank features of the covariance matrices in 8.7. In 8.9 and 8.10, and 8.11 we discuss the training setup used in our experiments and a link to the MKOR repository . In 8.12 we analyze MKOR and other optimizers on the training tasks as non-convex optimizers, only concerning their performance on training tasks. We conclude this section by proofs of the lemmas used in the paper.

## 8.1 ResNet-50

We train ResNet-50, a convolutional neural network with more than 25M parameters, on ImageNet, a text classification task with more than 1.2M samples. The same setup is used in [21], and SGD is used as the first-order baseline.

The target accuracy in this experiment is 75.9%. MKOR converges to this target accuracy in 57 epochs, while SGD, the first-order baseline, achieves this accuracy in 88 epochs. MKOR achieves $1.49\times$ speedup over SGD. KAISA, the second-order baseline converges in 54 epochs, but due to its expensive steps, MKOR still converges $1.04\times$ faster than KAISA. We do not compare to HyLo because HyLo is not able to achieve the target accuracy for ResNet (it reaches 75.6% with tuning as reported in [17] and our experiments confirm it). As shown, the effect of complexity reduction and improvement in performance in MKOR is less obvious in ResNet because the model dimension ($d$) is smaller compared to LLMs such as BERT. Please see Table 1 for comparison of complexity between methods.

We could not reproduce the ResNet-50 results of Eva [33] on ImageNet because the hyperparameters are not reported. We tried to tune Eva on multiple settings and none converged to desired accuracy. It is important to note Eva is not comparing results with the most efficient implementation of KFAC. The KFAC version used in Eva is from [16], dated to 2015. A number of followup works, mentioned in 1, have provided faster implementations of KFAC. We use KAISA [21], the state-of-the-art implementation of KFAC. Also from discussions with KAISA authors and our own experiments the optimal inversion frequency for KFAC is 200. Eva uses an inversion frequency of 50 for KFAC, which makes KFAC slower.

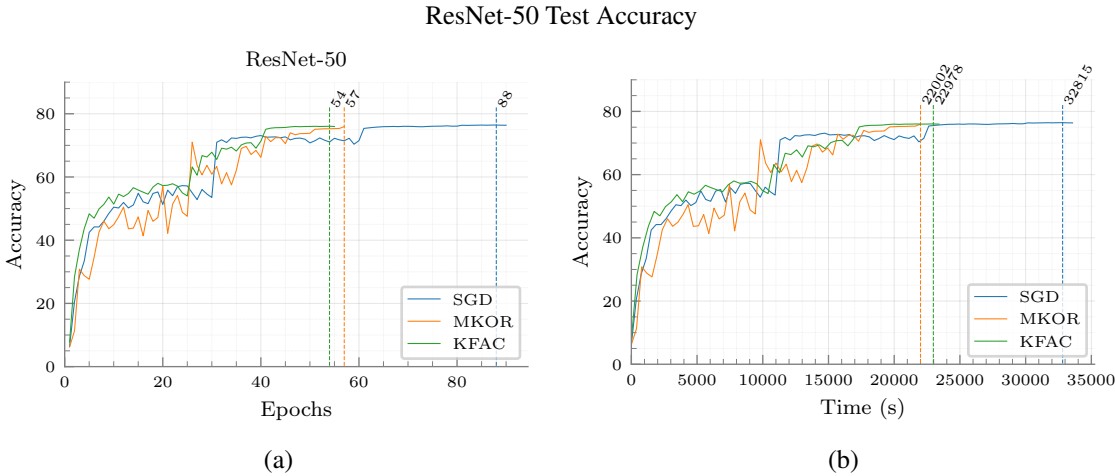

Figure 6: Test accuracy of ResNet-50 on ImageNet for MKOR, KAISA, and SGD on 64 GPUs.

Table 4: BERT-Large-Uncased Results on the GLUE classification tasks.

| Optimizer | Iterati-ons | MNLI (acc) | QQP (F1) | QNLI (acc) | SST-2 (acc) | COLA (mcc) | STS-B (corr) | MRPC (F1) | RTE (acc) | Avera-ge |
|---|---|---|---|---|---|---|---|---|---|---|
| LAMB | 1,563 | 0.841 | 0.878 | 0.913 | 0.919 | 0.516 | 0.875 | 0.812 | 0.664 | 0.8023 |
| KAISA | 1,563 | 0.821 | 0.854 | 0.900 | 0.921 | 0.489 | 0.878 | 0.888 | 0.617 | 0.796 |
| MKOR | 1,500 | 0.844 | 0.879 | 0.916 | 0.923 | 0.523 | 0.892 | 0.905 | 0.690 | 0.8214 |
| MKOR | 600 | 0.833 | 0.878 | 0.904 | 0.921 | 0.494 | 0.886 | 0.893 | 0.653 | 0.8078 |
| MKOR-H | 600 | 0.838 | 0.877 | 0.911 | 0.921 | 0.502 | 0.886 | 0.898 | 0.657 | 0.811 |
| Eva | 1000 | 0.839 | 0.877 | 0.907 | 0.914 | 0.499 | 0.890 | 0.904 | 0.650 | 0.809 |

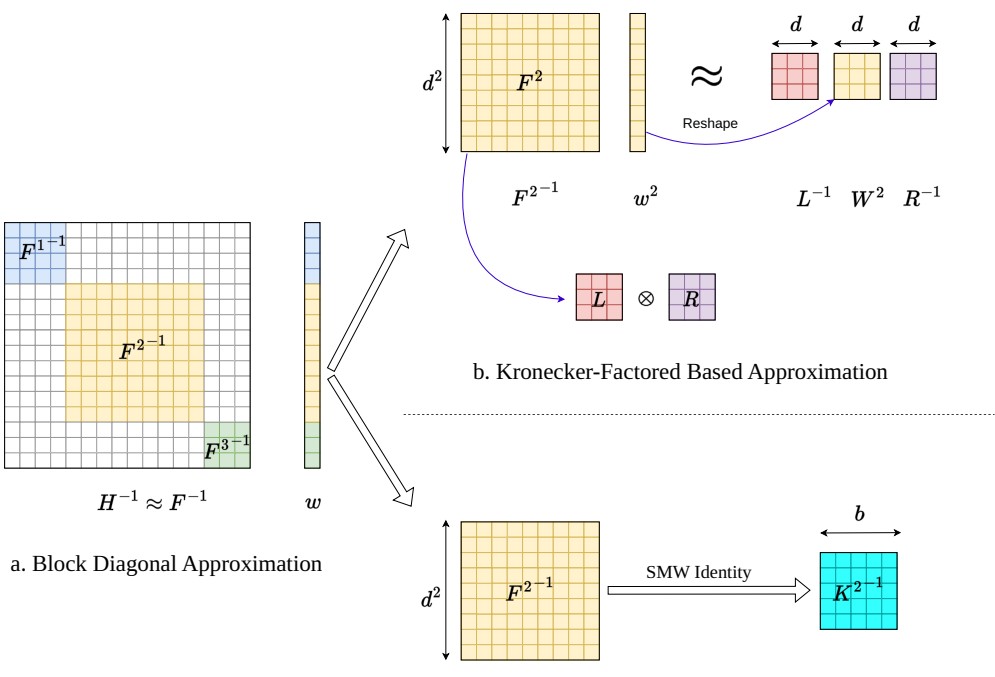

Figure 7: Approximations in second-order methods.

## 8.2 GLUE Results

We discussed the speedup achieved using MKOR on the GLUE dataset in Section 4. For completeness, Table 4 shows the metrics achieved in each of the different GLUE tasks on BERT-Large-Uncased trained on different optimizers.

## 8.3 Derivation of NGD Approximations

***Natural Gradient Descent (NGD)*** In NGD, which is a second-order method, we use the inverse of the Fisher Information Matrix (FIM) as a preconditioner to the gradients as shown in Figure 7-a. Equation 1 shows the update rule of NGD , where $F^m$ is the FIM block corresponding to block $m$. Equation 10 shows the definition of FIM for an arbitrary layer in our model, where $x_i$ is the $i^{th}$ sample in the batch.

$$F^m = \frac{1}{b} \sum_{i=1}^{b} \nabla_{x_i} \ell(\mathcal{W}, x_i) \nabla_{x_i} \ell(\mathcal{W}, x_i)^T \tag{10}$$

***Kronecker Factorization (KFAC).*** KFAC methods reformulate the FIM block as the Kronecker product of two matrices as shown in Equation 11 where $g^m = \nabla_{x^m}\mathcal{L}$ and $a^m$ is the vector form of the activation output of layer $m$ and $\mathbb{E}$ is the expectation operator and $x^m$ is the input matrix of layer $m$. Please note that we have used the mixed-product property of Kronecker multiplication for getting the right hand value.

$$F^m = \mathbb{E}[(g^m \otimes a^{m-1})(g^m \otimes a^{m-1})^T] = \mathbb{E}[(g^m g^{mT}) \otimes (a^{m-1}a^{m-1^T})] \tag{11}$$

Furthermore, we assume that $\mathbb{E}[(g^m g^{mT}) \otimes (a^{m-1}a^{m-1^T})] \approx \mathbb{E}[g^m g^{mT}] \otimes \mathbb{E}[a^{m-1}a^{m-1^T}]$, which is a strong assumption, but helps us simplify the computation further. Using the inversion property of Kronecker multiplication, we can compute the inverse of FIM using Equation 12.

$$F^{m-1}w^m = \mathbb{E}[g^m g^{mT}]^{-1} \otimes \mathbb{E}[a^{m-1}a^{m-1^T}]^{-1}w^m \tag{12}$$

By using the mixed Kronecker matrix-vector product property, we can get the update value in Equation 2, which is illustrated in Figure 7-b. We refer to $L^m$ and $R^m$ as the left and right factors respectively. Adding momentum to the left and right factors and denoting the iteration number with a subscript to the factors, we will get equations 3 and 4.

*Sherman-Morrison-Woodbury-Based Natural Gradient Descent (SNGD).* In this method, the SMW identity is used for approximating the inverse of $(F^m + \mu I) \in \mathbb{R}^{d^2 \times d^2}$, where $\mu$ is a damping factor used in the preconditioning. Equation 13 shows the process of computing the inverse of the FIM for a single layer in the network, where $A^m \in \mathbb{R}^{d \times b}$ is the batch of activations of layer $l$ and $G^m \in \mathbb{R}^{d \times b}$ is the batch of gradients of the loss function with respect to the inputs of that layer and $U = [\nabla_{W^m}\mathcal{L}(\mathcal{W}, x_1), ..., \nabla_{W^m}\mathcal{L}(\mathcal{W}, x_b)]^T \in \mathbb{R}^{d^2 \times b}$ is the concatenation of the gradients of the loss function with respect to the parameters of that layer and $\odot$ shows the Hadamard element-wise product. In this method, a kernel matrix in $\mathbb{R}^{b \times b}$ is inverted, as shown in Figure 7-c.

$$(F^m + \mu I)^{-1} = \frac{1}{\mu}(I - U^m(A^{m-1^T}A^{m-1} \odot G^{mT}G^m + \mu I)^{-1}U^{mT}) \tag{13}$$

## 8.4 Numerical Instability of Second-order Methods

In Section 3.3, we discussed that in second-order methods, multiple matrix inversion or root-finding algorithms need to be executed, which make the second-order methods prone to numerical instabilities. Furthermore, we discussed that left and right factors in second-order methods have large condition numbers, resulting in further issues in inversion. Figure 8 shows the eigenvalues of the right factor and its condition number for ResNet-50 model on CIFAR-10 dataset on KFAC algorithm. Even when using damping factors and filtering out extremely small eigenvalues, the condition number of these matrices is large, motivating for using double precision computations for avoiding numerical instabilities.

MKOR, on the other hand, doesn't suffer numerical instabilities when inverting such matrices, and its computational complexity isn't dependent on the condition number either.

## 8.5 Sensitivity to Learning Rate

Learning rate is one of the main hyperparameters in machine learning (ML) that can directly affect the convergence time of optimization, and ML practitioners have to spend a lot of time tuning this hyperparameter. More specifically, in first-order methods, a large learning rate can easily lead to divergence and numerical instability, and in second-order methods large learning rates can lead to exploding gradient descent as discussed in Section 3.3. Using small learning rates can lead to slow convergence in both first- and second-order methods, and can even jeopardize the main advantage of second-order methods, which is their faster convergence rate.

One of the main advantages of our method is its robustness against a wide range of learning rates. As Table 5 shows, first-order methods are extremely sensitive to learning rate, and the second-order methods are prone to ripples and divergent for a larger range of learning rates, and lose their performance for small learning rates. Our method, on the other hand, will converge with a high

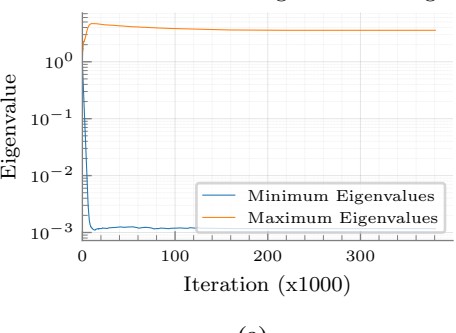

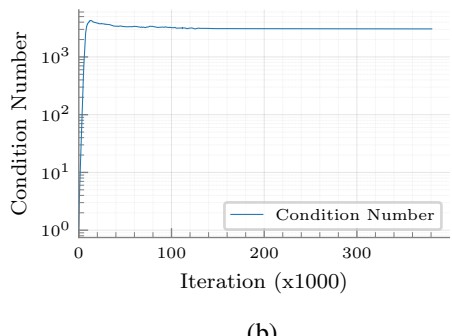

(a)                    (b)

Figure 8: Maximum and minimum eigenvalues (**a**) and the condition number (**b**) of the right factors in KFAC when training ResNet-50 on CIFAR-10. As illustrated, the minimum eigenvalues of the factors in KFAC approach zero, meaning that the factors are singular, and hence have large condition numbers, making numerical inversion of them complex and numerically unstable.

Table 5: Number of epochs necessary for convergence in different optimizers for ResNet-50 on CIFAR10. MKOR is the least sensitive optimizer to learning rate, converging in almost the same number of iterations for a wide range of learning rate, while other optimizers either diverge (**D**) or converge to a local-minimum (∗ **superscript**).

| Learning Rate 
 Optimizer | 10 | 1 | 0.1 | 0.01 |
|---|---|---|---|---|
| MKOR | 94 | 79 | 78 | 76 |
| KAISA | 112 | 100 | 90 | 89* |
| HyLo | D | 123* | 98 | 150* |
| SGD | D | D | 108 | 145* |

convergence rate for a wide range of learning rates, and by directly modifying the inverse of factors as discussed in 3.3 can find a proper equilibrium between first- and second-order methods. This table shows that our method is the least sensitive to the learning rate values and can make the job of ML practitioners for tuning this hyperparameter extremely easy.

## 8.6 Scalability

Figure 9 shows the strong scalability of MKOR on BERT-Large-Uncased on up to 64 GPUs.

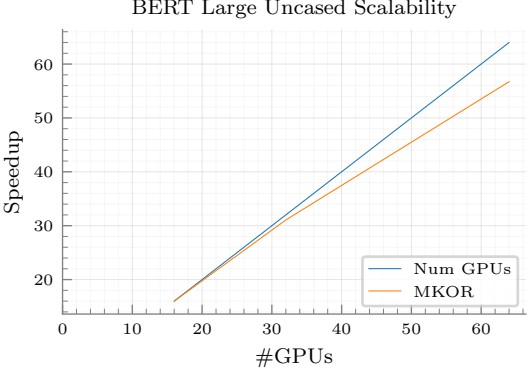

Figure 9: Scalability of MKOR.

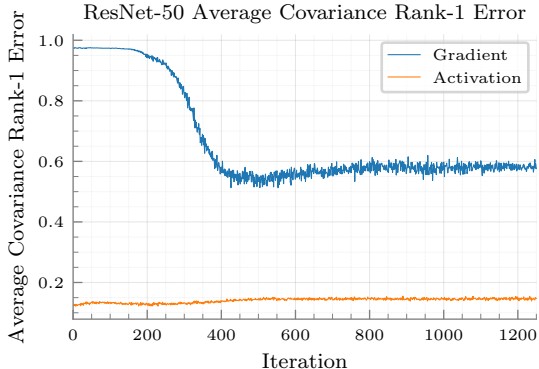

Figure 10: Average covariance rank-1 approximation error for ResNet-50 in different iterations

Table 6: Per-GPU memory usage (in GB) for MKOR, KFAC/KAISA, LAMB, and SGD on BERT-Large-Uncased pre-trainingResNet-50 training on ImageNet.

| Model | MKOR | KFAC/KAISA | LAMB | SGD |
|---|---|---|---|---|
| ResNet-50 | 3.88 | 5.83 | - | 3.01 |
| BERT | 23.34 | 29.97 | 12.80 | - |

## 8.7 Decaying Eigenvalues and Rank-1 Approximations

Figure 10 shows that the eigenvalues of the factors will decay as the model converges, making rank-1 approximations more effective. The reason behind the decay in the eigenvalues is that the weights are initialized randomly and the neurons work independently in the beginning of the training, but as the model converges, the neurons become more dependent on each other and thus the activation and input gradients will become linearly dependent. This is also reflected by some large error values in the distributions in Figure 5[a, b, c, d]. The factors in MKOR are initialized with identity, starting MKOR from a first-order method. As a result, MKOR is more robust against noise in the approximations in the first iterations (the approximation error does not noticeably affect the factors when replacing $L_{t-1}^{m-1}{}^{-1}$ and $R_{t-1}^{m}{}^{-1}$ in equations 5 and 6 with identity). But as the model converges, the factors in MKOR will be mostly shaped by the training samples, making MKOR more reliable on less erroneous approximations, and the decaying eigenvalues of the factors help MKOR with that.

## 8.8 Memory Overheads

The memory overheads of MKOR in comparison to other optimizers are reported in Table 6. It can be observed that all the second-order methods have significant memory overheads to the first-order methods, but MKOR reduces the overhead of KFAC/KAISA up to $1.5\times$.

## 8.9 Hyperparameters

**BERT-Large-Uncased.** For the BERT-Large-Uncased pre-training, we use the same hyperparameters used in [18]. The factors in KAISA are updated every 50 iterations, and the factors in MKOR and MKOR-H are updated every 10 iterations.

**ResNet-50.** For SGD and KAISA, we use the same hyperparameters used in [21]. The factors in MKOR are updated every 10 iterations, and the learning rate used there is the same is KAISA. The learning rate in MKOR decays by a factor of 2 at the end of epochs 25, 35, 40, 45, 50, 55, and 56.

## 8.10 Experiment Settings

For the BERT-Large-Uncased pre-training and fine-tuning experiments, we have used up to 64 A100 GPUs on the Polaris [3] cluster which has 560 nodes, each with 4 NVIDIA A-100 GPUs with NVLink interconnects.

Table 7: List properties of the models, datasets, and settings used in our experiments.

| Model | | Dataset | | | GPU | |
|---|---|---|---|---|---|---|
| Name | #Parameters | Name | Train | Test | Arch | # |
| BERT-Large-Uncased | 335.1M | Wikipedia - BookCorpus | - | - | A100 | 64 |
| ResNet-50 | 25.5M | ImageNet | 1.2M | 50k | V100 | 64 |
| AlexNet | 20.3M | CIFAR-100 | 50K | 10K | V100 | 4 |
| BERT-Base-Cased | 108.9M | SQuAD v1.1 | 87.6K | 10.6K | V100 | 4 |
| BERT-Large-Cased | 335.1M | IMDB | 25K | 25K | V100 | 4 |

The rest of the experiments are conducted on the Mist cluster [6], which has 54 nodes, each with 32 IBM power9 cores with 256GB RAM and 4 NVIDIA V100 GPUs with 32GB memory and NVLink inter-node connections and InfiBand EDF intra-node connections.

Each of the training experiments are conducted 5 times, and their median timing is reported. The reported accuracies are the median of the results we have obtained from multiple runs. For the scalability experiments, we have trained BERT-Large-Uncased for 100 iterations on two separate jobs for each setting and averaged them. For the time breakdown experiment, we have reported the median of 5 experiments.

Table 7 summarizes the models, datasets, and GPU architectures used in our training and fine-tuning experiments. In addition, the results in Figures 3 and 4-a are gathered on a single 4-GPU node of V100s. The results of Figure 4-b are gathered on 64 A100 GPUs.

## 8.11 Reproduction

Our code base is publicly available on `https://github.com/Mohammad-Mozaffari/mkor`, and the instructions for running each experiment are available there.

## 8.12 Training Accuracy Experiments

To evaluate MKOR as an optimizer that tries to minimize a specific objective function, we have considered the case of only minimizing the loss function of models on different tasks and set all the other optimizer parameters such as weight decay to zero. Since using a non-zero weight decay adds a quadratic term to the loss function and using different weight decays for different optimizers leads to optimizing different objective functions using different optimizers which might be considered unfair.

Recent work has shown the advantage of second-order methods over their first-order counterparts on multiple CNN tasks [17; 21], such as residual networks [9]. In our training accuracy experiment, we use another CNN benchmark, *AlexNet* [12] with more than 20M parameters on *CIFAR-100* [11] consisting of 50K training and 10K validation images of 100 classes. Figures 11-a and 12-a show the convergence properties of different optimizers. MKOR is $1.26\times$, $1.31\times$, and $1.58\times$ faster than HyLo-KIS, SGD, and KAISA respectively. The reason for the low convergence speed of KAISA is that we needed to use small learning rates for avoiding exploding gradients in it, which has damaged its convergence rate.

*BERT* is a large language model with two variants, *BERT-Base* with more than 108M parameters and *Bert-Large* with more than 335M parameters. As shown in Figures 11-c and 12-c, we have fine-tuned *BERT-Large* on the *IMDB* dataset which is a text classification task with 25K training and 25K test samples. MKOR outperforms SGD and HyLo-KIS by a speedup factor of $1.22\times$ and $1.43\times$ respectively. We have also fine-tuned *BERT-Base* on *SQuAD* dataset, which is a question answering task with 87.6K training and 10.6K test samples. MKOR achieves $1.26\times$ and $1.56\times$ speedup over SGD and HyLo respectively. We couldn't find a learning rate with which KAISA would converge on any of our *BERT* experiments which are based on the HuggingFace implementation of BERT, and the reason for lack of convergence of KAISA is exploding gradients.

## 8.13 Knee-Point Learning Rate Scheduler

While using large learning rates is crucial for utilizing the higher convergence rate of second-order methods, it is necessary to reduce the convergence rate after some iterations so that the model

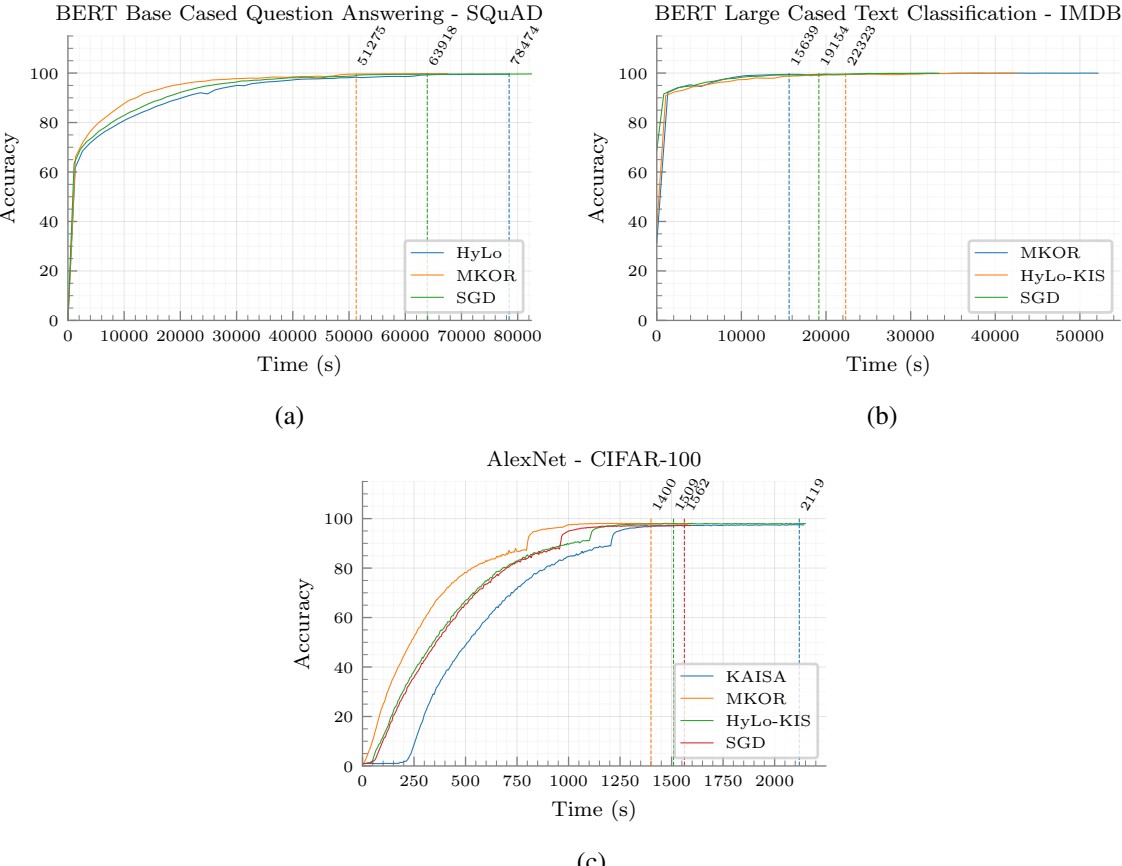

Figure 11: Training time for distributed first- and second-order optimizers SGD, MKOR, KAISA, and HyLo on *BERT-Large-Cased* on *IMDB* (**a**), *BERT-Base-Cased* on *SQuAD* (**b**), and *AlexNet* on *CIFAR-100* (**c**). In all the experiments, MKOR outperforms other optimizers in convergence speed.

converges, and the number of iterations for changing the learning rate is not known in advance. In practice, machine learning practitioners will manually find the number of iterations by trial and error or use predefined functions [10; 14] that don't necessarily work ideally in all optimization problems. We used knee-point learning rate scheduler in Section 8.12.

To fully utilize the potentials of the optimizers, we have designed a learning rate scheduler that monitors the rate of improvement in accuracy or decrease in the loss function value, and based on that decides when to decrease the learning rate. The scheduler detects knee-points in the accuracy/loss and decreases the learning rate when a knee-point is observed.

By definition, knee-points are defined as the points where the average accuracy/loss rate is less than $\beta$ times the increment/decrement in the accuracy/loss since using the current learning-rate. For averaging the accuracy/loss rate, we use an exponential moving average, and $\beta$ is a hyperparameter that we can choose to show how much the scheduler can tolerate lack of improvement to detect the accuracy/loss.

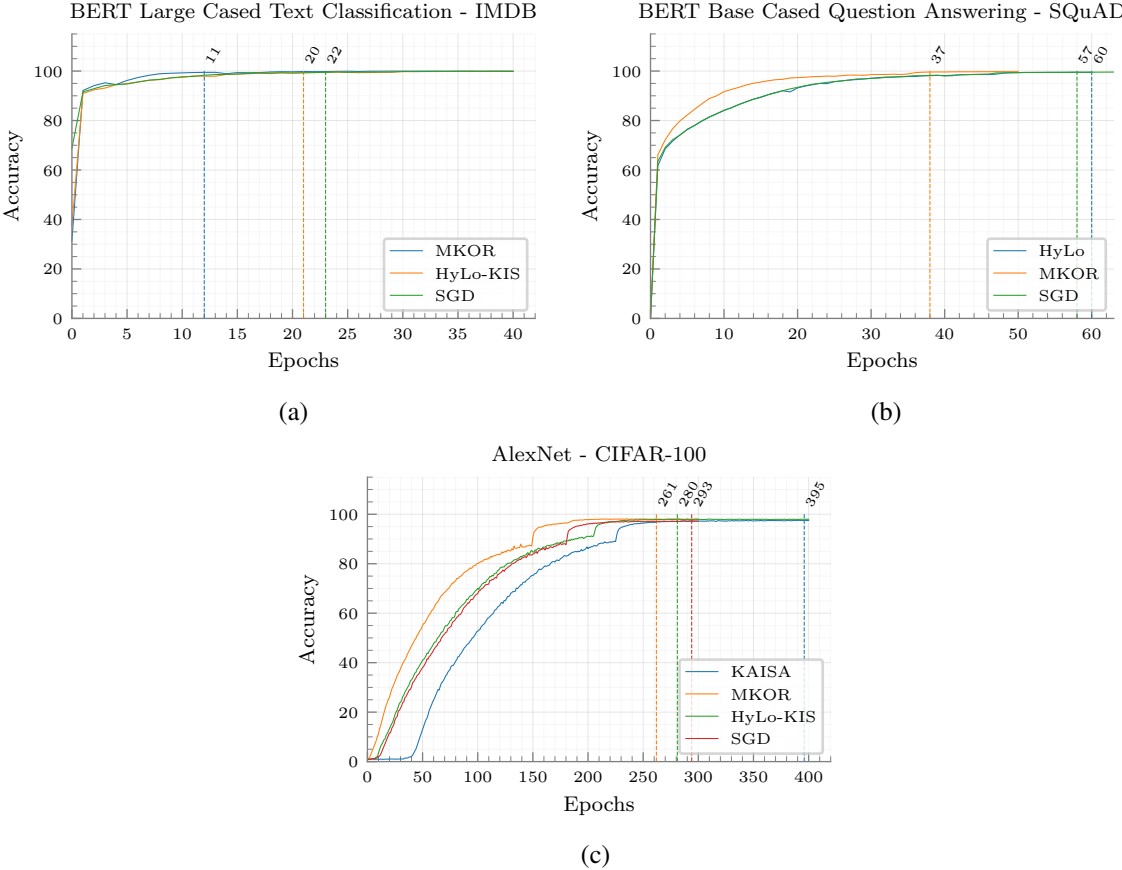

Figure 12: Training accuracy vs. the number of epochs for distributed first- and second-order optimizers SGD, MKOR, KAISA, and HyLo on *BERT-Large-Cased* on *IMDB* (**a**), *BERT-Base-Cased* on *SQuAD* (**b**), and *AlexNet* on *CIFAR-100* (**c**). In all the experiments, MKOR outperforms other optimizers in convergence rate.

## 8.14    Proofs

**Lemma 3.1**:

*Proof.* Given a matrix positive definite matrix $J_{t-1}$ and a vector $j$ and scalar $0 < \gamma < 1$, we show that Equation 14 results in a positive-definite matrix.

$$J_t^{-1} = \gamma J_{t-1}^{-1} + \frac{(1-\gamma)}{\gamma^2(1 + \gamma(1-\gamma)j^T J_{t-1}^{-1}j)} J_{t-1}^{-1} j j^T J_{t-1}^{-1} \tag{14}$$

Since $\gamma > 0$ and $J_{t-1}$ is a positive-definite matrix, $\gamma J_{t-1}$ is also positive-definite.

Also, since $J_{t-1}$ is a positive-definite matrix, $\forall x \neq 0 : x^T J_{t-1} x > 0$.

$$j^T L_{t-1}^{-1} j > 0 \xrightarrow{0 < \gamma < 1} 1 + \gamma(1-\gamma)j^T J_{t-1}^{-1} j > 0 \tag{15}$$

Now we show that $J_{t-1}^{-1} j j^T J_{t-1}^{-1}$ is also positive-definite.

$$\forall x \neq 0 : x^T J_{t-1}^{-1} j j^T J_{t-1}^{-1} x = (j^T J_{t-1}^{-1} x)^T (j^T J_{t-1}^{-1} x) = \|(j^T J_{t-1}^{-1} x)\|^2 > 0 \tag{16}$$

Since both matrices on the right-hand-side of Equation 14 are positive-definite and the sum of two positive-definite matrices is a positive-definite matrix, the left-hand-side of it will be also positive definite. $\qquad\square$

**Lemma 3.3**:

*Proof.* By defining $P = (\zeta L^{-1} + (1-\zeta)I) \otimes (\zeta R^{-1} + (1-\zeta)I)$, Equation 17 will hold.

$$\hat{\mathcal{L}}(w_0 - \Delta w) - \mathcal{L}(w_0) = -\nabla\mathcal{L}(w_0)^T P \nabla\mathcal{L}(w_0) \tag{17}$$

Now, we will show that $P$ is a positive-semi-definite matrix. By using the associativity property of Kronecker multiplication, we can represent $P$ as in Equation 18. Please note that different identity matrices in 18 have different shapes.

$$P = \zeta^2 L^{-1} \otimes R^{-1} + \zeta(1-\zeta)L^{-1} \otimes I + \zeta(1-\zeta)I \otimes R^{-1} + I \tag{18}$$

Since matrices $L$ and $R$ are positive-semi-definite, the Kronecker products $L^{-1} \otimes R^{-1}$, $L^{-1} \otimes I$, and $I \otimes R^{-1}$ are also positive-semi-definite. As a result, for any non-zero vector $x$, $x^T P x > 0$. So, based on Equation 17, $-\nabla\mathcal{L}(w_0)^T P \nabla\mathcal{L}(w_0)$. As a result, Equation 19 holds.

$$\hat{\mathcal{L}}(w_0 - \Delta w) < \mathcal{L}(w_0) \tag{19}$$

$\square$

## Lemma 3.2

*Proof.* Considering the quantization of matrix $J$ and vector $j$ in Equation 20 and assuming the maximum quantization error is $\epsilon$ and the maximum values in vector $j$ and matrix $J$ is $m$, we can consider one of the three possible cases:

1. *Vector-Vector Dot Product*: The resulting error is $O(2\epsilon m^2 d)$ because the error of each multiplication can at most be $2m\epsilon + \epsilon^2$ and by adding the $d$ multiplications, the maximum error can be $(2m+1)d\epsilon$.

2. *Vector-Matrix Product*: The resulting error is $O(2\epsilon m^2 d)$ because a vector-matrix product can be considered as multiple independent vector-vector dot products.

3. *Vector-Vector Outer Product*: The resulting error is $O(2\epsilon m)$ because each element in the resulting matrix is computed using a multiplication with the maximum error of $2\epsilon m^2 + \epsilon^2$.

$$\gamma J + \frac{(1-\gamma)}{\gamma^2(1 + \gamma(1-\gamma)j^T J j)} J j j^T J^T \tag{20}$$

The error imposed by quantizing $\gamma J$ is at most $\gamma\epsilon$. The quantization error in the denominator of the fraction in 20 is negligible in comparison to 1 and won't change the growth order of the final error term. The quantization error of $Jj$ is $O(2m^2 d\epsilon)$ as discussed earlier in the case of vector-matrix product. $J j j^T J^T = (Jj)(Jj)^T$ is a vector-vector outer product, resulting in $O(4m^3 d^2)$ error.

So the final error quantization error is $O((\gamma + 4\frac{(1-\gamma)}{\gamma^2}m^3 d^2)\epsilon)$

$\square$

