# OpenReview forum: "MKOR: Momentum-Enabled Kronecker-Factor-Based Optimizer Using Rank-1 Updates"
_NeurIPS.cc/2023/Conference — NeurIPS 2023 poster_

### Official Review · Reviewer_dQpV · 2023-06-30

**Soundness:** 2 fair
**Presentation:** 2 fair
**Contribution:** 2 fair
**Rating:** 5
**Confidence:** 4

**Summary:**

This work introduces a new optimization algorithm for deep neural networks. Building upon the baseline Kronecker-factored curvature (K-FAC) algorithm, this new approach approximates each activation and gradient covariance matrix within K-FAC as a rank-one matrix. This approximation enables the efficient calculation of the inverse Hessian. By leveraging this enhanced efficiency, the algorithm allows for more frequent updates of the Hessian. Experimental results demonstrate great speed improvements compared to the state-of-the-art first and second order optimization algorithms.

**Strengths:**

1. Efficient second-order optimization algorithm is important for the training of DNNs.

2. Experimental results show great speed-ups compared to the baseline algorithms.

3. The paper is in general well-written, and the main contribution is clear.

**Weaknesses:**

One notable weakness of this research lies in the insufficient justification of the proposed technique, which involves applying rank-1 approximation to the covariance matrices used in K-FAC. While the work addresses the efficiency benefits of this method, it fails to thoroughly discuss the aspect of why employing rank-1 approximation does not impact the final accuracy or convergence of the algorithm. The following questions should be addressed to strengthen the research: What specific scenarios allow for accurate rank-1 approximation, and when might it be ineffective? Does the effectiveness of the approximation depend on the singular value decays of the covariance matrices? Are there any illustrative toy examples that can shed light on these considerations?

**Questions:**

n/a

---

> ### Author Rebuttal · Authors · 2023-08-09
>
> In the below and the attached PDF figure-file we show that the covariance matrices in training neural networks can be approximated by rank-1 matrices both empirically and theoretically. We will include these discussions in our paper to address the reviewers’ concerns.
>
> **Experimental Results:** As shown in figure 2 in the attached PDF figure-file, our experiments show that the covariance matrices can be approximated with rank-1 matrices with low error and higher rank approximations are unnecessary in practice. Figure 2 (in the attached PDF figure-file) shows the error distribution of the optimal rank-1 approximation methods of the covariance matrices in ResNet-50 and BERT-Large-Uncased pretraining. Our extensive tests on well-known benchmarks (shown in the paper) show this property holds for all the models and we have not come across a benchmark that does not have low-rank covariance matrices.
>
> **Decaying Eigenvalues:** Furthermore, figure 2.e in the attached PDF figure-file shows that the eigenvalues of the factors will decay as the model converges, making rank-1 approximations more effective. The reason behind the decay in the eigenvalues is that the weights are initialized randomly and the neurons are working independently in the beginning of the training, but as the model converges, the neurons will be more dependent on each other and hence the activation and input gradients will become linearly dependent. This also is reflected by some large error values in the distributions in figure 3[a, b, c, d] in the attached PDF figure-file. The factors in MKOR are initialized with identity, starting MKOR from a first-order method. As a result, MKOR is more robust against noise in the approximations in the first iterations (the approximation error does  not noticeably affect the factors when replacing ${L_{t-1}^{m-1}}^{-1}$ and ${R_{t-1}^{m}}^{-1}$ in equations 5 and 6 with identity). But as the model converges, the factors in MKOR will be mostly shaped by the training samples, making MKOR more reliable on less erroneous approximations, and the decaying eigenvalues of the factors help MKOR with that.
>
> **Analysis:** Small batch sizes and overparameterization of the networks will lead to low-rank covariance matrices in deep neural networks. Let’s consider the covariance matrix $C=XX^T$, where $C \in R^{d \times d}$ is the covariance matrix and $X \in R^{d \times b}$ is a matrix in which each column corresponds to a single sample and $d$ and $b$ are the sample dimension and the per-GPU batch size respectively. Rank of the covariance matrix  is $min(b, d)$. In case the per-GPU batch sizes are small, the covariance matrices in each GPU will be significantly low-rank, and rank-1 approximation methods can work well in those scenarios. In case the batch sizes in each GPU are large, we observe that the covariance matrices will stay low-rank. The underlying reason for this observation is that current neural networks are over-parameterized, and as a result, different features in the covariance matrices of the activations and the output gradients won’t be linearly independent, resulting in low-rank covariance matrices.
>
> **Extending MKOR to Higher Ranks:** Furthermore, one can extend MKOR to use higher-rank covariance matrices. Let’s assume that $C = \sum_{i=1}^{r}{c_i c_i^T}$ where $r$ is the rank of the covariance matrix $C$. We can apply SMW identity to compute $C_1^{new} = (C^{old} + c_1 c_1^T)^{-1}$ with $O(d^2)$ computational complexity. Then we can compute $C_2^{new} = (C_1^{new} + c_2 c_2^T)^{-1}$ using SMW identity with $O(d^2)$ computational complexity. We can continue the same pattern by computing $C_i^{new} = (C_{i-1}^{new} + c_i c_i^T)^{-1}$. The total computation complexity of this process will be $O(rd^2)$. We should add this cost to the cost of computing the low-rank approximation of $C$ which requires an SVD. Using SVD kills the main advantage of using low-rank computations, since the computational complexity of applying SVD is the same as inverting the factors directly. We couldn’t find any cheaper way to compute low-rank approximations of the covariance matrices, except for the rank-1 approximation used in this paper.

---

> > ### Comment · Reviewer_dQpV · 2023-08-19
> >
> > Thank you for the response, it would be good to add the discussions above to the paper. I will raise my score.

---

### Official Review · Reviewer_zw2W · 2023-07-06

**Soundness:** 3 good
**Presentation:** 3 good
**Contribution:** 3 good
**Rating:** 7
**Confidence:** 4

**Summary:**

MKOR introduces Kronecker rank-1 representation of covariance for second-order optimizers. Authors try to solve the complexity of large language models in second-order optimizers while the Fisher information approximation is computed using rank-1 Kronecker matrix factorization. They also propose a hybrid method to cover the first-order method. Their evaluation of large language models is fair and sound.

**Strengths:**

The main idea behind this research is simple but solves the important problem of the applicability of second-order optimizers for large language models. The paper is well-motivated and experiments are sound and relevant.




**Weaknesses:**

- Authors only argue why rank-1 is proposed from a practical standpoint but there is no theoretical justification to support why rank-1 is enough for Fisher information approximation.


**Questions:**

- Rank1 approximation might be restrictive to approximate the Fisher information properly. I wonder in some cases an extension to higher ranks is necessary. I wonder if an extension towards SeKron [1] is natural.
[1] SeKron: A Decomposition Method Supporting Many Factorization Structures arXiv:2210.06299

- Does adding a learning rate scalar in Equation (2) make the method more flexible to converge?
- How well does the second-order MKOR method compare with the first-order with the same training budget in terms of time and memory?

**Limitations:**

The numerical stability requires SVD decomposition and matrix inversion which are expensive to compute.

---

> ### Author Rebuttal · Authors · 2023-08-09
>
> **Higher Rank Approximations:**
>
> As shown in figure 2 in the attached PDF figure-file, our experiments show that the covariance matrices can be approximated with rank-1 matrices with low error and higher rank approximations are unnecessary in practice. Figure 2 (in the attached PDF figure-file) shows the error distribution of the optimal rank-1 approximation methods of the covariance matrices in ResNet-50 and BERT-Large-Uncased pretraining. Our extensive tests on well-known benchmarks (shown in the paper) show this property holds for all the models and we have not come across a benchmark that does not have low-rank covariance matrices.
>
> *Sekron and other approximation methods:* MKOR  reduces the factor inversion costs in second-order methods by combining SMW identity with low-weight rank-1 approximation methods. If we use any other method except for rank-1 approximations, e.g. Kronecker approximation discussed in SeKron, we need to find proper formulations to incorporate them in the matrix inversion. To our knowledge, no such cheap formulations exist that can be used for inverting matrices.
>
> It might be possible to extend MKOR to use higher-rank covariance matrices however it will introduce significant computational overheads to MKOR since it requires computing the SVD of the factors. Let’s assume that $C = \sum_{i=1}^{r}{c_i c_i^T}$ where $r$ is the rank of the covariance matrix $C$. We can apply SMW identity to compute $C_1^{new} = (C^{old} + c_1 c_1^T)^{-1}$ with $O(d^2)$ computational complexity. Then we can compute $C_2^{new} = (C_1^{new} + c_2 c_2^T)^{-1}$ using SMW identity with $O(d^2)$ computational complexity. We can continue the same pattern by computing $C_i^{new} = (C_{i-1}^{new} + c_i c_i^T)^{-1}$. The total computation complexity of this process will be $O(rd^2)$. We should add this cost to the cost of computing the low-rank approximation of $C$ which requires an SVD. Using SVD kills the main advantage of using low-rank computations, since the computational complexity of applying SVD is the same as inverting the factors directly. We couldn’t find a cheaper way to compute low-rank approximations of the covariance matrices, except for the rank-1 approximation used in this paper.
>
> **Learning Rate in MKOR:**
>
> Thanks to the reviewer’s note, we realized a typo in equation 2. The learning rate is indeed accounted for in our implementation of equation 2. Equation 2 should be $W^m := W^m - \alpha {(L_t^m)}^{-1} \nabla_{W^m} \mathcal{L} {(R_t^m)}^{-1}$ where $\alpha$ is the learning rate. We will fix this typo in the final version of the paper.
>
> **Comparison to First-Order Methods:**
>
> As shown in figure 2 and 6 in the paper (our appendix), MKOR-H converges 2.57$\times$ faster than LAMB (the state-of-the-art first-order optimizer for pretraining BERT-Large-Uncased) and 1.49$\times$ faster than SGD (the state-of-the-art first-order optimizer for training ResNet-50 on ImageNet). So given the same time budget, MKOR converges faster than other first-order methods. The memory overheads of MKOR in comparison to other optimizers are also reported in table 3 in the attached PDF figure-file. It can be observed that all the second-order methods have significant memory overheads to the first-order methods, but MKOR reduces the overhead of KFAC/KAISA up to 1.5$\times$. We will magnify these in the final version.
>
> **SVD in Low-Rank Approximations:**
>
> Instead of using SVD methods that suffer from numerical instability and high computational overheads, MKOR uses a low-weight averaging method to compute rank-1 approximations (lines 2 and 3 in Algorithm 1 and lines 106 to 109 in the paper text). Hence, MKOR is SVD-free and its computational and memory overheads are negligible.

---

> ### Comment · Reviewer_zw2W · 2023-08-18
> **Rating revised**
>
> I would like to thank the authors for their convincing rebuttal. I raise my rating accordingly.

---

### Official Review · Reviewer_Ar2i · 2023-07-07

**Soundness:** 4 excellent
**Presentation:** 4 excellent
**Contribution:** 3 good
**Rating:** 7
**Confidence:** 4

**Summary:**

The work introduces a second-order optimizer that uses rank 1 covariance activation and gradient statistics, and efficient inversion algorithms to accelerate an approximated estimate of 2nd order information. Another addition is the rescaling of gradients and norm-based stabilization, which help to stabilize the entire optimization procedure. This procedure yields a highly stable 2nd order optimizer that is both memory (communication) and computationally efficient and leads to improved convergence times compared to baselines.


**Strengths:**

From the work its clear that the method depends on many independent factors. While some reviewers might see this as a weakness, I see this is a strength. Developing efficient 2nd order optimizer is a highly technical and difficult undertaking and this work provides multiple components that make a stable optimizer possible that is both memory and computationally efficient.

The evaluation could have been extended to regular language modeling since BERT pretraining has become less academically relevant. Nevertheless, a BERT large pretraining is a rather convincing experimental setup. As such, the experimental robustness is also a strength of the paper.

**Weaknesses:**

I think additional experiments for causal language modeling would have been beneficial since this would be the main advantage for more efficient training (BERT pretraining is fast enough for most researchers, but most researchers are unable to pretrain LLMs due to their computational cost).

While this additional experiment would greatly improve the paper, the experimental methodology in this paper is already relatively robust.

**Questions:**

No questions in particular.

**Limitations:**

Limitations are not discussed.

---

> ### Author Rebuttal · Authors · 2023-08-09
>
> MKOR is currently the most efficient second-order optimizer that can be used on LLMs and CNNs. As the reviewer has pointed out, similar to most state-of-the-art optimizers, MKOR has hyperparameters that users have to tune to use it properly. We have tried to automate as many of these tunings as possible, e.g. the use of norm-based stabilizer for automating stabilization or the stabilization frequency for switching between first- and second-order methods in MKOR-H or knee-point learning rate scheduler. We are optimistic on MKOR’s performance on other LLMs however were not able to try other models due to their computational and resource costs.

---

### Official Review · Reviewer_KDBs · 2023-07-10

**Soundness:** 2 fair
**Presentation:** 2 fair
**Contribution:** 2 fair
**Rating:** 5
**Confidence:** 4

**Summary:**

The paper proposes a second-order optimizer named MKOR based on K-FAC. Compared with K-FAC, MKOR uses a rank-1 update to construct Kronecker factors enabling the inverse of Kronecker factors to be efficiently computed via the Sherman-Morrison-Woodbury (SMW) identity. In addition, the rank-1 update of factors in MKOR also helps reduce the communication traffic in distributed training with data parallel for aggregating factors. Experiments on the BERT-Large-Uncased model with the GLUE dataset show that MKOR outperforms first-order LAMB and second-order KAISA.

**Strengths:**

1. An implicit inverse computation method with the SMW identity in computing the inverse preconditioner factors and a formula to control the balance of exploitation between the first-order and second-order information.
2. A hybrid method of MKOR with the first-order method, MKOR-H, which allows switchable from MKOR to the first-order optimizer in the later of training.

**Weaknesses:**

1. An important baseline, Eva [A], which is extremely similar to MKOR, is missing.
2. Though MKOR-H seems to achieve some improvement, when switching MKOR to the first-order optimizer is unclear.
3. The comparison between MKOR and K-FAC seems to be not that fair (e.g., Figure 4(b)).

[A] Eva: Practical Second-order Optimization with Kronecker-vectorized Approximation, ICLR 2023.

**Questions:**

1. The key idea of using rank-1 updates to replace the original Kronecker factors in K-FAC has been proposed in Eva [A]. In [A], the authors provide a better derivation of the preconditioner without storing the factor matrices so that they use Kronecker vectors (KVs) to calculate the inverse of KVs using the SMW formula, which is much better than MKOR as MKOR needs to store the matrix form of factors (i.e., $L_t^m$ and $R_t^m$). The similarity between MKOR and Eva and the advantage of Eva over MKOR make the originality of this work weak. It is suggested to clarify the differences between MKOR and Eva, and highlight some advantages of MKOR over Eva.
2. The main novel idea in MKOR is the balance between second-order and first-order information in updating $L_t^m$ and $R_t^m$ using Eq. (7) and (8) by controlling $\zeta$. However, how to choose $\zeta$ (or dynamically changing $\zeta$ to switch MKOR to SGD just like MKOR-H) is very important. The paper doesn’t provide such an analysis of when and how to use $\zeta$ though some results show that it can help improve convergence.
3. Similarly to the previous problem, MKOR-H is a combination between first-order optimizer and MKOR according to the loss during training. However, how to measure whether the loss is changed is unclear (e.g., results in Figure 2 haven’t such information either) and when to switch to the first-order optimizer is unclear.
4. What are the update frequencies of the second-order information for MKOR and K-FAC in Figure 2 and Table 3? And how do you choose such frequencies for comparison?
5. In the original paper of KAISA, the update frequency can be set to be very high (e.g., 100-500) for achieving good accuracy on ResNet models with CIFAR10 and ImageNet datasets, is it possible that MKOR could be better than KAISA in these kinds of scenarios?

[A] Eva: Practical Second-order Optimization with Kronecker-vectorized Approximation, ICLR 2023.

**Limitations:**

See Questions.

---

> ### Author Rebuttal · Authors · 2023-08-09
>
> **Comparison to Eva:**
>
> What the reviewer has pointed out as the strengths of Eva (using KVs to not store the factor matrices), is at the same time a weakness for Eva, leading Eva to being less accurate and less efficient vs. MKOR. Using KVs instead of factor matrices disables the proper use of momentums and imposes the use of damping factors in Eva, both of which damage the accuracy of FIM approximations.
>
> Similar to KFAC, EVA’s formulation involves a damping factor (which helps with numerical stability similar to the left-hand-side of equation 13 in the appendix) that introduces additional error to the FIM approximation, however in MKOR we introduce a new formulation that removes the damping factor by storing the inverse of the previous factors and utilizing the SMW identity (equations 5 and 6 in our paper).
>
> In addition, Eva doesn’t support proper momentum-based training because it doesn’t store the left and right factors of the FIM (but instead averages the KVs in time which does not have theoretical backup). MKOR stores the left and right factors to enable momentum in training, which has been shown to significantly boost the training process (equations 3 and 4 in our paper).
>
> MKOR has two additional contributions that lead to a more stable and efficient implementation of rank-1 updates. (1) It uses a *Norm-Based Stabilizer* and *Gradient Rescaling Mechanism* to detect and avoid exploding gradients. (2) with MKOR-H we combine the high convergence rate of second-order methods in the initial phase of training with the low overhead of first-order methods in the late stages of the training.
>
> Eva comparison Results (please also see the attached PDF figure-file):
>
> * BERT-Large-Uncased: Compared to Eva, MKOR-H is faster in pretraining BERT-Large-Uncased by a factor of 1.69. Its accuracy is also better (an average accuracy of 81.1\% on GLUE and 90.64\% on SQuAD v.1 for MKOR with only 600 iterations vs 80.9\% on GLUE and 90.55\% on SQuAD v.1 for EVA with 1000 iterations), please see Figure 1 and Table 1 and Table 2 in the attached PDF figure-file. Figure 1 in the PDF file shows the convergence results of all the optimizers used in the paper in addition to Eva.
>
> * ResNet-50: We could not reproduce the ResNet-50 results of Eva on ImageNet because the hyperparameters are not reported. We tried to tune Eva on multiple settings and none converged to desired accuracy.
>
> * It is important to note EVA is not comparing results with the most efficient implementation of KFAC. The KFAC version used in EVA is from [1], from 2015. A number of followup work (please see line 41-42 in the paper) have provided faster implementations of KFAC. We use KAISA, the state-of-the-art implementation of KFAC. Also from discussions with KAISA authors and our own experiments the optimal inversion frequency for KFAC is 200, Eva uses an inversion frequency of 50 for KFAC, which makes KFAC slower.
>
> **Dynamic Controlling of $\zeta$:**
>
> Changing the stabilization frequency with a constant $\zeta$ (as done in MKOR) is equivalent to dynamically controlling the value of $\zeta$. In MKOR, instead of changing $\zeta$ dynamically for each iteration, we keep it constant but we change the frequency that we stabilize the factors (equations 7 and 8 in the paper). The frequency is dynamically controlled using a norm-based criteria (lines 5 and 6 in Algorithm 1). Also, if the inversion frequency is higher than a threshold, which is similar to having a small small $\zeta$, we switch to SGD to save on computations.
>
> **Difference of MKOR and MKOR-H in Loss and Switching Criteria:**
>
> MKOR combines the first- and second-order optimization methods using the norm-based stabilization. Per our response to the previous question, the frequency of the stabilization dictates how close MKOR is to first-order methods, i.e. the more frequent the stabilization takes place, the closer MKOR gets to first-order methods. Our switching method looks at the average stabilization frequency in the last few iterations using an exponential moving average, and if the frequency is higher than a threshold and the loss change rate is less than a small ratio of the overall loss reduction, the user will be notified to switch to LAMB (the first-order optimizer) if needed. In the BERT-Large-Uncased pretraining, we switch to LAMB on iteration 300.
>
> **Inversion Frequency**
>
> As reported in section 7.6 in the appendix of the paper, in all our experiments, the frequency (factor reuse time) of inverting the factors in MKOR and MKOR-H is set to every 10 iterations unless stated otherwise (i.e. in Figure 4 in the paper). For KFAC we use the default settings they provide in their papers.
>
> **Infrequent Factor Inversion in KAISA:**
>
> For our experiments, including the ResNet-50 experiment, we used the default settings in the KAISA paper, which has a factor reuse time of 200. This number can be large in KAISA, since the factor inversions in KAISA are computed from scratch every time. In MKOR, which is an approximation-based method, we cannot use very stale factors, since each of our updates only modifies the factors slightly and stale factors won’t be useful anymore.
>
> **Fairness of Figure 4.b Comparisons:**
>
> Figure 4.b motivates the use of more frequent updates in second-order information, and it does not intend to compare different optimizers. There might be some confusion when interpreting Figure 4, particularly due to the absence of MKOR 1000 and KAISA 10. Kaisa 10 is not in the figure because it was very slow in comparison to others (we will add a sentence in the figure caption to note this), MKOR 1000 updates the factors so infrequently with very insignificant rank-1 updates that it performs similar to SGD and won’t have any benefit compared to other second-order methods.
>
> [1] Optimizing neural networks with kronecker-factored approximate curvature, Martens et. al., 2015
>
> [2] KAISA: An Adaptive Second-Order Optimizer Framework for Deep Neural Networks, Paulostki et. al., 2021

---

> > ### Comment · Reviewer_KDBs · 2023-08-18
> > **Rebuttal read**
> >
> > Acknowledgments to the authors for providing a detailed response and additional experiments. Most of my concerns are addressed so I raise my rating. Hope that these discussions would be included in the final version to make the paper clear.

---

### Author Rebuttal · Authors · 2023-08-09

We thank all reviewers for their very informative feedback. We have provided a separate answer to each reviewer and have also attached a PDF file with figures and tables that add to our rebuttal (which we refer to as the attached PDF figure-file in our per-reviewer responses).

---

### Decision · Program_Chairs · 2023-09-21

**Decision:**

Accept (poster)

**Comment:**

The submission presents a family of heuristics for K-FAC which improve its memory and computational complexities and allow adaptively falling to first-order optimization. Experiments on a medium-scale setup show promising accuracy and overall performance.

The reviewers found the author discussion convicing enough to converge to an acceptance decision. The authors are encouraged to include the discussion points into their next revision.